# Application of C-InGAN Model in Interpretable Feature of Bearing Fault Diagnosis

**DOI:** 10.3390/e26060480

**Published:** 2024-05-31

**Authors:** Wanyi Yang, Tao Liang, Jianxin Tan, Yanwei Jing, Liangnian Lv

**Affiliations:** 1School of Artificial Intelligence, Hebei University of Technology, Tianjin 300130, China; 202232804028@stu.hebut.edu.cn; 2Hebei Jiantou New Energy Co., Ltd., Shijiazhuang 050018, China; 3Goldwind Science & Technology Co., Ltd., Wulumuqi 830063, China

**Keywords:** bearing fault diagnosis, interpretable feature fault diagnosis, C-InGAN, attention

## Abstract

Although traditional fault diagnosis methods are proficient in extracting signal features, their diagnostic interpretability remains challenging. Consequently, this article proposes a conditionally interpretable generative adversarial network (C-InGAN) model for the interpretable feature fault diagnosis of bearings. Initially, the vibration signal is denoised and transformed into a frequency domain signal. The model consists of the two primary networks, each employing a convolutional layer and an attention module, generator (G) and discriminator (D), respectively. Latent code was incorporated into G to constrain the generated samples, and a discriminant layer was added to D to identify the interpretable features. During training, the two networks were alternately trained, and the feature mapping relationship of the pre-normalized encoder was learned by maximizing the information from the latent code and the discriminative result. The encoding that represents specific features in the vibration signal was extracted from the random noise. Ultimately, after completing adversarial learning, G is capable of generating a simulated signal of the specified feature, and D can assess the interpretable features in the vibration signal. The effectiveness of the model is validated through three typical experimental cases. This method effectively separates the discrete and continuous feature coding in the signal.

## 1. Introduction

Rolling bearings in rotating machinery, such as that of a wind turbine, typically operate under complex environmental conditions. Such complex operating and environmental conditions result in higher failure rates when compared with other components, thereby increasing operational and maintenance costs. Consequently, it is essential to develop an effective method for bearing fault diagnosis in order to ensure the smooth operation of rolling bearings, particularly when maintaining large-scale equipment such as wind turbines [1,2,3].

Traditionally, bearing fault diagnosis has primarily relied on signal processing methods in order to analyze the vibration signals linked to bearing faults. Expert knowledge is then utilized to extract fault features, establishing the foundation for fault diagnosis. Techniques including wavelet transform, empirical mode decomposition (EMD), complete ensemble empirical mode decomposition with adaptive noise (CEEMDAN), and variational mode decomposition (VMD) were employed [4,5,6,7,8]. Despite the effectiveness of these traditional techniques, they all share a common drawback: they require significant expertise in the design and fine-tuning of the feature extraction process, making them less accessible for non-experts and challenging to automate.

In contrast, deep learning techniques represent a paradigm shift in fault diagnosis. Leveraging data-driven approaches, deep learning algorithms such as convolutional neural networks (CNNs) and recurrent neural networks (RNNs) can automatically learn to capture complex features from raw data, significantly reducing the reliance on extensive expert knowledge. These models have the advantage of being able to learn directly from data, identifying intricate patterns and anomalies that might be overlooked by human analysts or traditional methods [9]. To address the challenges of gradient vanishing and explosion during vibration signal data training, Wang et al. [10] utilized a long short-term memory network (LSTM) for gearbox fault diagnosis. Furthermore, Wang et al. [11] introduced a deep separable convolutional network for fault diagnosis. Detection data from various sensors are directly fed into the neural network model for fault diagnosis. Liu et al. [12] incorporated an attention module into a one-dimensional convolutional neural network (1D-CNN). This enhanced model allocates weights to different feature channels via the attention module, thus enhancing the accuracy of fault diagnosis. Zhang et al. [13] introduced a multi-mode CNN (MMCNN), which extracts features through multiple parallel convolutional layers, thus amplifying the comprehensiveness and effectiveness of the extracted feature information. The gray-scale image produced by the continuous wavelet transform (CWT) is then fed into the MMCNN for training, enhancing the fault diagnosis accuracy under varying working conditions. Although these fault diagnosis methods effectively distinguish between different fault types, their application is limited to identifying individual faults. However, the ‘black box’ nature of deep learning models can be a drawback, as it obscures the decision-making process, making it difficult to interpret the results. This limitation is critical in industrial applications where understanding the cause and nature of a fault is as important as detecting its presence.

A generative adversarial network (GAN) is used to produce imitation samples [14]. Since its introduction in 2014, GAN has been extensively applied to computer vision and various other domains. The GAN is capable of learning the distribution of the training data and producing corresponding imitation samples. This inspiration has led numerous researchers to utilize GAN for the fault diagnosis of rolling bearings. Cao et al. [15] developed an adversarial signal generative adversarial network (AdvSGAN) to create simulated vibration signals, aiding training with real data. The experimental findings demonstrate that the fault diagnosis model is reliable when processing vibration signal data with an elevated signal-to-noise ratio. W. Wan et al. [16] augmented the GAN’s network architecture by integrating a self-attention layer into both the generator and discriminator. This enhancement improved the networks’ feature extraction capabilities and produced superior imitation samples in order to train the fault diagnoser, consequently increasing the diagnostic accuracy. Nevertheless, GAN-generated samples frequently exhibit data imbalance and possess limited diversity. Peng et al. [17] developed a generative framework based on Wasserstein conditional GANs, amalgamating Wasserstein loss with hierarchical feature matching loss to bolster the model’s generalization capacity. Yang et al. [18] converted the bearing’s vibration signal into a grayscale image and employed a convolutional layer for image feature extraction. By integrating this approach with a conditional generative adversarial network (CGAN), they created imitation samples to address the problem of inadequate data samples. These studies leverage GAN and its variants to produce imitation samples, aiding in the training of fault diagnosers and mitigating challenges, such as inadequate sample data and unbalanced fault types. However, these methodologies depend on the GAN for data augmentation, necessitating supplementary diagnostic training for fault diagnosis, thereby incurring substantial computational expenses.

Bearing vibration signals exhibit nonlinear and non-stationary characteristics. Before processing, the fault features in these signals are typically embedded in a high-dimensional surface space with an irregular arrangement. When processed using traditional methods, vibration signals result in points of the same feature clustering together in space; however, the extracted feature relationships are disordered, in contrast with the desired clear and interpretable feature relationships [19]. Hence, this paper proposes an interpretable feature fault diagnosis method, termed the conditionally interpretable generative adversarial network-based fault diagnosis method (C-InGAN). This method reduces the dimensionality of state points, originally embedded in a high-dimensional surface space, to the visual space, based on their flow patterns. This allows features of the original coupling interference, such as working condition, fault type, and damage size, to be decoupled and sequentially organized according to the coordinate axis, making the coordinate axis interpretable. Figure 1 illustrates these interpretable feature relationships.

This paper’s primary contributions are as follows:It proposes a C-InGAN model for diagnosing interpretable features of bearings. This entails decoupling the initially coupled features and aligning them along the coordinate axis, which is then interpreted.In the proposed C-InGAN model, the generator aids in training using real data by creating simulated samples during the training phase. Concurrently, the trained discriminator in the model directly serves as a fault diagnosis tool, eliminating the need for further training of the diagnosis network.This study incorporated an attention mechanism into the discriminator’s network architecture. The mechanism assigns differential weights to various network channels based on their importance, thereby boosting the network’s capability for feature extraction.

The remainder of this paper is organized as follows. The second part offers a comprehensive introduction to the interpretable fault diagnosis model. The third part validates the method’s effectiveness of the method using three representative experimental data sets. The final section summarizes the study and presents the conclusions.

## 2. Interpretable Bearing Fault Diagnosis Model

To address the challenge of limited interpretability in traditional feature extraction methods for fault diagnosis, this paper introduces a C-InGAN model tailored for the interpretable feature fault diagnosis of bearings. The detailed processing flow is shown in Figure 2. The steps are outlined as follows:Vibration signal acquisition: Bearing vibration signals that were acquired under various operational conditions were gathered and normalization was conducted.Vibration signal preprocessing: The acquired signals were decomposed using VMD with optimal parameters. Then, the most pertinent intrinsic mode functions (IMFs) were selected for signal reconstruction based on the principle of correlation [20]. Subsequently, the reconstructed signals underwent a fast Fourier transform (FFT), with the frequency domain signals serving as the model’s input. Data were partitioned into training and testing sets.Model training: The C-InGAN model was constructed, the training set data fed into it, and adversarial learning employed in order to alternately fine-tune the model’s generator and discriminator until the iteration concludes.Interpretable fault diagnosis: The trained C-InGAN model discriminator was deployed as a fault classifier to execute interpretable fault diagnosis on the test set data.

### 2.1. Data Preprocessing

The initial vibration signal typically contains considerable noise, which can obscure fault-related information. Therefore, applying preprocessing methods to the vibration signal is advantageous when enhancing the fault diagnosis accuracy. These preprocessing techniques include data regularization, VMD decomposition for denoising, and time–frequency transformation.

Data regularization: The data were appropriately segmented based on different datasets. In this paper, the training and test sets were divided in an 8:2 ratio, employing stratified sampling to categorize various fault data from the dataset. The collected data sometimes deviate from the center value, presenting challenges in subsequent time–frequency processing. Therefore, the original data were regularized. The specific procedure is outlined as follows:(1)$${x_{k}}^{\prime }=x_{k}-\frac{1}{n}\sum_{i=1}^{n}x_{i}$$
where ${x_{k}}^{\prime }$, $x_{k}$, and $\frac{1}{n}\sum_{i=1}^{n}x_{i}$ represent the transformed data point, current data point, and average value of the whole data, respectively.

VMD decomposition denoising: Variational mode decomposition (VMD) is a signal processing method that can adaptively match the optimal center frequency and limited bandwidth of each mode to achieve the effective separation of intrinsic mode components (IMFs) [4]. The bearing vibration signal was decomposed using VMD. Several IMFs, exhibiting the strongest correlation with the original signal, were chosen for reconstruction in order to eliminate extraneous components and simplify fault information extraction.

Time–frequency transform processing: The frequency domain signals are more effective in highlighting the differences between various fault signals compared with the time domain signal. As a result, the denoised signals are subjected to an FFT, with subsequent processing based on the frequency domain signal obtained from this transformation.

### 2.2. C-InGAN Model

A generative adversarial network (GAN) is a deep learning model that combines two networks: a generator (G) and a discriminator (D) [14]. The generator (G) produces pseudo-samples with maximum realism through training and learning, while the discriminator (D) assesses the authenticity of the input samples to the best extent possible. Throughout the training process, G and D underwent gradual optimization as they engaged in mutual confrontation. GAN training continues until the discriminator (D) is unable to distinguish between the true and false samples. The loss function for the GAN is as follows:(2)$$\underset{G}{\min}\;\underset{D}{\max}V\left(G,D\right)=E_{x∼P_{data}}\left[\log D\left(x\right)\right]+E_{z∼P_{z}}\left[\log\left(1-D\left(G\left(z\right)\right)\right)\right]$$
where G and D stand for the generator and the discriminator, respectively. E denotes the expectation operation; x and z stand for real data and random noise, respectively; and $P_{data}$ and $P_{z}$ stand for the distributions of real data and random noise, respectively.

However, the data generated by the original GAN were not constrained, and the discriminator could only identify the authenticity of the data. The features learned by GAN are mixed and encoded in a complex and disorderly manner in the data space. InfoGAN adds mutual information constraints based on GAN, making parts of the disordered feature coding meaningful, such as the category and skewness of the numbers in the MNIST dataset. These characteristics incorporate both discrete and continuous features.

Building on this concept, we introduce the conditionally interpretable generative adversarial networks (C-InGAN). In the generator, a portion of the random noise is transformed into latent code. The latent code is a fundamental concept in GAN, representing various data features. Specifically, the latent code is a set of latent variables derived from a portion of random noise. In the generator, the latent code constrains the generated output, enabling the generated samples to exhibit specific characteristics. For the discriminator, the output not only determines the authenticity of the data, but also distinguishes various types of interpretable data characteristics, which may be either discrete or continuous. G and D underwent alternate training, with the parameters of one fixed while training the other. During the training process, the model learns the mapping relationship between the generator’s input latent code and discriminator’s output discriminant result by maximizing the information from both the input latent code and the discriminator’s output. Through training iterations, the generator can produce increasingly realistic samples constrained by the latent code, exhibiting specified characteristics. The discriminator is tasked not only with identifying the data’s authenticity but also with differentiating the data’s features, becoming more accurate as training iterations advance. Compared with the original GAN, the new adversarial network adds a constraint condition for the information between the latent code and generated data discrimination result based on its loss function. The loss function is as follows:(3)$$\underset{G}{\min}\;\underset{D}{\max}V\left(G,D\right)=E_{x∼P_{data}}\left[\log D\left(x\right)\right]+E_{z∼P_{z}}\left[\log\left(1-D\left(G\left(z\right)\right)\right)\right]-\lambda I\left(c;D\left(G\left(z,c\right)\right)\right)$$
where λ and c stand for the regularization parameter and latent code, respectively. I(c;D(G(z,c))) represents the mutual information between the latent code c and the generated data D. Mutual information quantifies the dependency between two random variables. In this study, the neural estimation method is employed. The parameter λ determines the weight of information between the latent code and the discriminative results of the generated data. A larger value of λ emphasizes the interpretability of the generated samples, while a smaller value focuses more on the authenticity of the generated samples. In this study, we selected λ=0.1 based on the results of a series of preliminary experiments which indicate that this value effectively balances the authenticity and interpretability of the generated samples. Additionally, the dimensionality of the latent code determines the extent of features the model can capture and generate. Higher dimensionality allows the representation of more complex features but also increases the model’s complexity and training difficulty. In this study, we selected a latent code dimensionality of 100, as this dimension strikes a balance between the model’s complexity and the quality of the generated samples in practical applications. Through the selection of these parameters, the C-InGAN model demonstrates excellent performance in balancing the authenticity and interpretability of the generated samples.

### 2.3. Network Structure of the Model

The C-InGAN model proposed in this paper includes two networks: a generator and a discriminator. Both network structures are based on a 1D-CNN. A squeeze-and-excitation network (SE-net) was introduced into the discriminator to enhance feature extraction results. The SE-net adds an attention module in the channel dimension, focusing on key operations of squeeze and excitation. Through automatic learning, that is, using another new neural network, the significance of each channel in the feature map is determined, and this importance is used to assign a weight value to each feature, enabling the neural network to concentrate on specific feature channels [21].

The specific description of the model network structure is shown in Figure 3.

The input of the generator was divided into two parts: random noise and latent code. The length of the random noise was 100, following a uniform distribution ranging from −1 to 1. The latent code comprises a concatenated combination of one-hot codes, each of which represents a sample label. The output was a synthesized one-dimensional artificial signal. The generator’s intermediate layer consisted of four deconvolution layers and two fully connected layers. The activation function of the deconvolution layer was the rectified linear unit (ReLU) function, and batch normalization was performed after each deconvolution operation.

For the discriminator, a one-dimensional real signal or false signal is taken as the input. There are three outputs in total: one output determines whether the signal is true, the activation function of the output layer is a sigmoid function with a length of 1. The remaining two outputs are based on additional fully connected networks, which are used to judge the discrete interpretable features of signals. For discrete features, such as fault type, the output layer’s activation function is the Softmax function, with its length corresponding to the number of discrete features. For continuous features, such as speed, the output layer does not have an activation function and has a length of 1. The intermediate layer consists of four convolution layers and pooling layers. The convolution operation involves one large convolution kernel and three smaller convolution kernels. The convolution layer employs the ReLU activation function, and SE-net is then added to weight different features. Finally, two pooling layers were added to reduce the feature dimension.

### 2.4. Model Training Process

The preprocessed frequency domain signals were fed into the proposed C-InGAN model for training. The generator learns the feature distribution of the real data and then generates data with features similar to those of the real data. It adheres to the prior conditions specified by the latent code in the generator’s input. The latent code represents the type of feature present in the signal. The generator produces imitation samples in accordance with this feature. The discriminator not only identifies the authenticity of the input data, but also distinguishes the feature types of the input data as a classifier, regardless of whether the input discriminator is true or false. During the C-InGAN network training, G and D undergo alternating training. D is trained initially, followed by G. Specifically, it is divided into two processes, as shown in Figure 2.

During the training of the discriminator, the network parameters of the generator were kept constant. Random noise Z and latent code C were fed into the generator to generate a synthetic sample adhering to the specified feature C. True and false samples were mixed into the discriminator, and the labels of the calibration true and false samples were 1 and 0. The feature label of the calibration true sample is a one-hot encoding of its true feature, and the label of the calibration false sample is the latent code C input by the generator. The loss function for this process comprises the following two components:(4)Loss=GAN_Loss+C_Loss
where GAN_Loss denotes the loss when determining whether the samples are true. C_Loss denotes loss when determining the features of the samples. In cases where there is more than one feature, this can also be divided into GAN_Loss and C_Loss. The network parameters of the discriminator are adjusted according to the total loss. The ultimate objective is to make the discriminator not only judge the authenticity of the input sample, but also distinguish the features of the input sample.

Conversely, when training the generator, the network parameters of the fixed discriminator remain unchanged, and random noise Z and latent code C are also input into the generator to produce false samples. In this process, the true sample is not needed, and the false sample is only input into the discriminator. Unlike the previous process, the label for false samples was set to 1 in this case. This is because the objective of this training phase is to maximize the deception of the discriminator using the samples generated from the generator. Similarly, the label for false samples corresponds with the latent code C input by the generator. The loss function of this process is the same as in the above formula; however, because of the change in the calibration label, the training target also changes accordingly. The ultimate goal of adjusting the network parameters of the generator according to the loss is to enable the generator to generate false samples as accurately as possible. It strives to meet the prior conditions specified by the latent code in the generator’s input. It is worth noting that, because feature C encompasses both discrete and continuous features, it corresponds to categorical cross entropy and MSE, respectively. Meanwhile, it is utilized to differentiate between true and false, representing a binary classification problem, and is consistently set to binary cross entropy.

The two training processes are alternated because the training goals of these processes are contradictory and mutually antagonistic. Therefore, with the progress of this training method, the two networks progress toward each other, and the performance of the generator and the discriminator is also improved. Consequently, the model’s data generation capacity, the discrimination capability for true and false data, and discernment of data features were all enhanced. At the end of the iteration, the network training was completed.

### 2.5. Innovation

In this paper, a C-InGAN model is proposed for bearing interpretable fault diagnosis. Unlike other methods, which only examine individual features in signal data, this approach assesses data based on two-dimensional feature quantities and organizes them into spatial coordinates following specific rules. This method can discern interpretable features based on their spatial location.

Unlike other GAN models employed in bearing fault diagnosis, the primary goal of their training is to acquire a sample generator. This generator is utilized to augment the training data, thereby addressing the inherent data challenges. Subsequently, additional diagnostic networks were trained to accomplish fault diagnosis. The training generator in the proposed C-InGAN, unlike other models, was designed to aid the discriminator in completing the training more effectively. Specifically, the convergence speed of the discriminator is significantly improved. For example, the discriminator achieves stable convergence within 50 epochs, whereas traditional models typically require around 100 epochs. The ultimate objective was to train the discriminator to directly accomplish fault diagnosis.

The SE-net was incorporated into the network structure of the discriminator. Through training, the significance of each channel to the outcome was determined, assigning an appropriate weight value to each channel and thereby enhancing the network’s feature extraction capability.

## 3. Case Studies

In this section, three experimental cases are used to demonstrate the effectiveness of the proposed method. These include the bearing dataset from Case Western Reserve University (CWRU), the time-varying speed bearing dataset from the University of Ottawa, and the accelerated degradation bearing dataset from Xi’an Jiaotong University. The operating environment of this experiment is an Intel i5-9300H (Intel, Santa Clara, CA, USA), NVIDIA Gtx1660Ti (Nvidia, Santa Clara, CA, USA), and the running memory of the device is 16 G.

### 3.1. Case 1: CRWU Data

#### 3.1.1. Data Description and Preprocessing

A bearing fault dataset from Case Western Reserve University (CWRU) was employed to validate the effectiveness of this method when extracting discrete feature quantities. The experimental bearing used is the SKF6205 (SKF, Gothenburg, Sweden), which was subjected to various fault types and degrees. The bearing vibration signal was captured using an acceleration sensor. The selected data include a 0 hp load and a speed of 1797 rpm, with a sampling frequency of 12,000 Hz [22]. The specific descriptions are listed in Table 1.

Data were sampled using windowing and were divided into 150 samples, with each sample being 2000 points in length. The data were stratified, with 80% of each dataset sampled as the training set, amounting to 1440 samples, and the remaining 20%, or 360 samples, forming the test set.

The data underwent regularization and VMD decomposition processing, and the results are displayed in Figure 4. The five IMFs most strongly correlated with the original signal were selected to reconstruct the signal, whereas those with less information were eliminated. The reconstructed signals were transformed into frequency domain signals by FFT, and the signal length was 1000. As shown in Figure 5, the VMD decomposition results of a 0.07-inch damage degree outer ring fault bearing signal show that the processed frequency domain signal contains less irrelevant information and more pronounced features.

As Figure 6 shows, in the schematic diagram of the 12 different fault signals after preprocessing, the discrimination between different signals is clearer.

#### 3.1.2. Experiment and Result Analysis

The processed data were used for the training of the C-InGAN model, and the category labels of the samples were set, being the fault location label c1 and the damage size label c2, respectively. There are four kinds of c1 labels, represented by 0~3 and five kinds of c2 labels, represented by 0~4. This specific situation, where the labels were converted into one-hot encoding, is shown in Table 1. The activation functions of the two feature outputs form the Softmax function.

In the training of the model, the Adam algorithm was used to adjust the network weight, with a batch size of 64, a learning rate of 2 × 10^−4^, and the number of iterations set to 10,000. Upon completion of the iterations, the test set samples were fed into the trained discriminator to distinguish the fault type and degree, with the discriminant results displayed as a visual coordinate map. The results are presented in the form of a visual coordinate map. As shown in Figure 7, the visual results are shown as three-dimensional maps. The remaining three maps are the main, side and top views of the three-dimensional map, which represent the classification results under different feature types. Firstly, (a) shows an overview of the three-dimensional diagram of the test set diagnosis. The three axes of xyz represent the fault location, the degree of damage, and the number of samples, respectively. Then, (b) represents the classification of the fault location, while (c) is the classification of the degree of damage. Finally, (d) provides an interpretable diagnosis of the two discrete features: fault location and degree of damage.

The test results misjudge the characteristics of only two datasets, where bearings with 0.028-inch ball faults were identified as 0.021-inch ball faults, yielding an accuracy rate of 99.4%. Figure 8a shows the experimental effect diagram obtained according to the method in [23]. The process begins with signal treatment using VMD, followed by selecting specific IMF components based on the Holder coefficient, and then extracting a multi-feature combination that includes Renyi entropy, singular value, and Hjorth parameter of the component. Finally, the t-SNE was used for dimensionality reduction visualization to obtain the resulting diagram. It can be seen that the 12 samples can be well distinguished, but the vector distribution does not have interpretable significance. Figure 8b shows the result of interpretable fault diagnosis. The abscissa axis calibrates the fault position of the bearing samples, and the ordinate axis calibrates the fault degree of the bearing sample. The fault features are arranged according to law.

### 3.2. Case 2: Ottawa Variable Speed Bearing Dataset

#### 3.2.1. Data Description and Preprocessing

The time-varying speed bearing vibration dataset from the University of Ottawa was utilized to validate the effectiveness of this method when extracting continuous feature quantities. A bER16K ball bearing, equipped with an accelerometer, was used to collect the bearing vibration data. The dataset comprised two experimental settings: fault type and variable speed conditions. The bearing fault types include normal, inner ring failure, and outer ring failure. The working speed conditions included increasing speed, decreasing speed, increasing then decreasing speed, and decreasing then increasing speed. The sampling frequency was 200,000 Hz and the sampling time was 10 s [24]. A detailed description is shown in Table 2.

Each dataset was divided into 1000 samples, each 20,000 points long, with 80% of each dataset sampled as the training set. Four training sets and one test set were derived from every five groups of data to maintain the continuity of the rotational speed characteristics of the test data. The four rotational speed conditions included increasing speed, decreasing speed, increasing then decreasing speed, and decreasing then increasing speed. Under each rotational speed condition, there were 3000 samples, comprising 2400 in the training set and 600 in the test set. The total amount of data is the sum of the four rotational speed conditions; 12,000 samples and 9600 samples are training samples. A total of 2400 samples were used as test samples.

Data regularization and VMD decomposition processing. The five IMFs with the strongest correlation to the original signal were also calculated to reconstruct the signals, eliminating IMFs with less information. The reconstructed signals were transformed into a frequency domain signal by FFT, and the signal length was 10,000. Figure 9 shows the VMD decomposition results of the inner circle fault bearing signal. Figure 10 shows a comparison of the data obtained before and after processing.

The graphs of the 100th, 300th, 500th, 700th and 900th sets of data of the three sets of acceleration signals were taken as the above preprocessing. It can be considered that the five sets of signals represent the data of the five speed segments from low to high speed, such as in Figure 11. The three sets of signals all show obvious regular changes when the speed increases, and the discrimination between signals at different fault positions is more obvious. Because the high-frequency signal shows little discrimination, the part before 20,000 Hz of the frequency domain signals is intercepted.

#### 3.2.2. Experiment and Result Analysis

The processed data were utilized to train the C-InGAN model, with the fault type label c1 and speed label c2 established. There are three kinds of c1 labels, which are represented by 0~2 and converted into one-hot encoding. The length of the c2 label was set to 1, corresponding to the calibration of the bearing’s speed. The size was 9.8~28.9, as shown in Table 2. The output activation function for the fault location feature is the Softmax function, while the speed feature’s output activation function is not specified.

Similarly, the model employed the Adam algorithm with a batch size of 64, a learning rate of 2 × 10^−4^, and, because of the larger dataset, the number of iterations was set to 15,000. After the training was completed, the test set was input into the trained discriminator to distinguish the fault type and fault degree of the test data. The results are displayed in three-dimensional graphics. Figure 12, Figure 13, Figure 14 and Figure 15 represent the four speed conditions of increasing speed, decreasing speed, increasing then decreasing speed, and decreasing then increasing speed. Each set of graphs shows the diagnosis results of 600 test data under this speed condition. In Figure 12a represents the overview effect of the three-dimensional graph. The three axes of xyz represent the samples, speed and fault location respectively. Figure 12b represents the classification of the three fault locations and ‘○’, ‘□’ and ‘*’ represent the three fault locations of inner circle, outer circle and normal, respectively. Figure 12c shows the diagnostic error display for speed, where the line formed by black points in the figure calibrates the actual speed. It can be seen that the speed of diagnosis is basically consistent with the calibrated speed label, and the error is small. Figure 12d shows the interpretable diagnostic result of speed, and the color of the sample points changed from blue to red, indicating that the speed changes from low to high. The accuracy of the fault type in the test sample was 100%. The first 200 samples in each group are inner circle faults, the middle 201–400 are outer circle faults, and the last 200 data are normal signals. The model could accurately determine the speed of the test signals.

In Figure 16a shows the experimental effect diagram according to the traditional signal processing method. Vibration data were adopted under the condition of increasing speed. The processing method involves initially treating the signal with VMD, selecting specific IMF components based on their correlation, and extracting the Renyi entropy of these components. This is followed by combining singular values and Hjorth parameters, with the final result (a) obtained through t-SNE dimensionality reduction visualization. The color of the data points changes from blue to red, and the speed of the data points changes from low to high. This method can be seen to a certain extent. The speed change of the bearing occurs at three fault locations, but the arrangement is irregular. Figure 16b shows the interpretable diagnostic result obtained by the proposed method. The ordinate axis calibrates the fault location of the bearing. IA, OA, and HA represent the inner circle, outer circle fault, and normal bearing acceleration data, respectively. The abscissa axis calibrates the speed of the data. The color of the data points from blue to red indicates that the speed judged by the model is from low to high. The data points were arranged according to the law of coordinate calibration. The data points were systematically arranged in accordance with the coordinate calibration.

### 3.3. Case 3: Xi’an Jiaotong University Accelerated Degradation Bearing Data Set

#### 3.3.1. Data Description and Preprocessing

The Xi’an Jiaotong University accelerated degradation bearing data set is used to verify the extraction effect of this method on the health degree of continuous change features [25]. The test bearing of the data set is an LDK UER204 rolling bearing. The degradation process of the bearing is accelerated by maintaining a fixed radial force and a high speed load, so as to obtain the full life cycle monitoring data of the test bearing. Three sections of bearing data were selected for experiments. Three bearings were tested under three working conditions, and the final bearing damage position was an outer circle fault. The experimental sampling frequency was 25.6 kHz, sampling 1.28 s every 1 min, 32,768 points each time, and 30,000 points before each sampling were intercepted. This was divided into three health segments with a length of 10,000, with the first 6000 points of each health segment used as experimental data. Each bearing had a data volume of 3 times the number of sampling times. The data order was disordered according to the 8:2 division of training set and test set data, with a training set of 955 data and a test set of 239 data. A detailed description of the data is shown in Table 3, where c1 is the condition label and c2 is the health label.

Data regularization and VMD decomposition processing. The five IMFs with the strongest correlation to the original signal were also calculated to reconstruct the signal, eliminating IMFs with less information. The reconstructed signals were transformed into frequency domain signals using FFT, and the signal length was 3000. Figure 17 shows the VMD decomposition result of the bearing signal under the first working condition. Figure 18 shows comparison of time domain and frequency domain signals before and after processing.

Considering the data of the 10th, 70th, 150th, 220th, 290th and 360th sections of the bearing data under the first working condition, it can be considered that this group of data represents the six bearing health sections from low to high, which are recorded as health sections 1 to 6. The frequency domain diagrams of the six health sections are shown in Figure 19. It can be observed that the frequency domain signal of the data changes regularly when the health degree increases. Therefore, using the frequency domain signal as experimental data is justified.

#### 3.3.2. Experiment and Result Analysis

The processed data are used for training the C-InGAN model, and the category labels of the samples are set as the working condition label c1 and the health label c2, respectively. There are three kinds of c1 labels, and 0~2 represent working conditions 1, 2 and 3, which are converted into one-hot codes. The length of the c2 label is set to 1, c2∈(0,1], and the health of the bearing data was calibrated. The health changes were linear. The output activation function of the working condition feature was set as the Softmax function, and the health feature output was the sigmoid function.

In the model training, the Adam algorithm was used to adjust the network weight. The batch size was set to 64, the learning rate was set to 2 × 10^−4^, and the number of iterations was set to 15,000. After the iterations were completed, the test set was input into the trained discriminator to differentiate the working conditions and health status of the test data, and the results are displayed using three-dimensional graphics. There are 239 data points in the test set in Figure 20, and there are 74, 97 and 68 test data points, respectively. The points of the three different shapes in the legend represent different working conditions of the bearing data. The color of the scale on the right side of the figure transitions from blue to red, indicating a gradual change in data health from low to high. In the figure, the health of the data points can be judged by color, or by the scale of the coordinate axis calibration. Figure 20a represents an overview of the three-dimensional graph. The three axes of xyz represent the samples, health degree and working conditions of the bearing respectively. Figure 20b shows the classification of the three working conditions of the test set data. Figure 20c displays the diagnostic results for the health degree feature. The line connected by the black points in the figure indicates the real health degree. It is evident that the diagnosed health degree is essentially consistent with the calibrated health label, with minimal error. Figure 20d presents the interpretable diagnostic results for the bearing data. The vertical axis calibrates the working conditions of the bearing data in the test set, and the horizontal axis represents the health of the data. The spatial positions of the sample points in the figure have interpretable significance. The test data results are entirely accurate, with an accuracy rate of 100%. Simultaneously, the model accurately assesses the health features of the test data.

In Figure 21a shows the vibration data under the three working conditions. The processing involves treating the signal with VMD, followed by extracting the singular value, sample entropy, and energy entropy of the components most strongly correlated with the original signal. The peak, mean, and kurtosis values of the reconstructed signal were calculated to create the final feature vector, with dimensionality reduction performed using t-SNE to produce (a). The points of different colors in the graph represent the bearing data under different working conditions. It can be seen that the three working conditions have good discrimination, and that the diagnostic accuracy of the working conditions is 99.8%. The diagram drawn by the arrow is the result of dimensionality reduction of bearing data under working condition 1. The health status of the data can be inferred from the color of its points, where blue to red indicates a change from a lower to a higher health status. However, the coordinate space of the data distribution lacks interpretative significance. Figure 21b displays the result obtained using the C-InGAN interpretable fault diagnosis model. The horizontal and vertical axes are interpretable, and the vertical axis calibrates the three working conditions of the bearing data. The health change of the horizontal axis calibration data, the color of the data points from blue to red indicates that the speed judged by the model is from small to large, and the data are arranged according to the law of coordinate calibration. The feature arrangement has interpretable significance, and the discrimination accuracy of the test data condition is higher, reaching 100%. Therefore, the C-InGAN interpretable fault diagnosis model can effectively extract the health characteristics of the bearing data well.

## 4. Conclusions

To address the issue of non-interpretability in fault diagnosis models, this paper proposes an interpretable feature fault diagnosis model based on C-InGAN. Firstly, the vibration signals are denoised and converted into frequency domain signals using a signal processing method. Subsequently, these frequency domain signals were input into the C-InGAN learning model for training. The network structure of the model was based on 1D-CNN, and an SE-net was introduced to improve the performance of network feature extraction. Finally, the discriminator that can output interpretative diagnostic results is obtained by training the model generator and discriminator. The model’s diagnostic effectiveness for both discrete and continuous features was verified using three datasets.

The contributions of this paper are as follows: A C-InGAN model is proposed to diagnose the interpretable features of bearings. The model can evaluate the feature quantities of data across multiple dimensions and systematically arrange them in spatial coordinates based on specific rules. The results are interpretable. The C-InGAN model used in this paper was not used to solve the problem of insufficient or unbalanced data, but to train the discriminator of the GAN model to directly realize fault diagnosis. SE-net is incorporated into the discriminator’s network structure, with a focus on different network channels during training, enhancing the network’s feature extraction capability. 

In the broader context of fault diagnosis, the implications of these results are profound. The ability of the C-InGAN model to accurately identify and classify various fault conditions with high reliability suggests its applicability across a wide range of industrial machinery. This capability could lead to significant improvements in predictive maintenance strategies, reducing downtime and maintenance costs while increasing operational efficiency.

This paper explores the application of an interpretable feature fault diagnosis model for fault type, speed and other characteristics, confirming its effectiveness. Despite the significant advantages demonstrated by the C-InGAN model, this study has several limitations, such as the biases that may be introduced during the dataset selection and preprocessing stages. The public datasets used in this study have specific conditions and environments, and the model’s generalization capability under different conditions has not yet been validated. The C-InGAN model has been primarily validated for bearing fault diagnosis, and its applicability to other types of mechanical equipment or industrial systems requires further investigation. This study assumes that the denoised vibration signals can adequately reflect fault characteristics; however, in practical applications, noise and other interferences might be more complex, potentially affecting the model’s diagnostic performance. In future work, we aim to incorporate improved network structures to develop adversarial models and endeavor to apply interpretable diagnosis to other significant research areas, such as predicting remaining useful life.

## Figures and Tables

**Figure 1 entropy-26-00480-f001:**
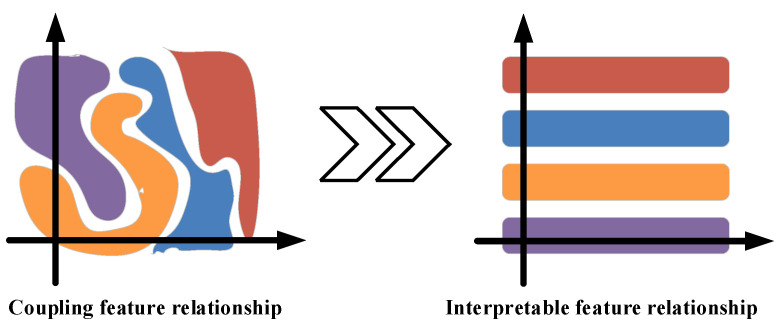
Interpretable feature relationship.

**Figure 2 entropy-26-00480-f002:**
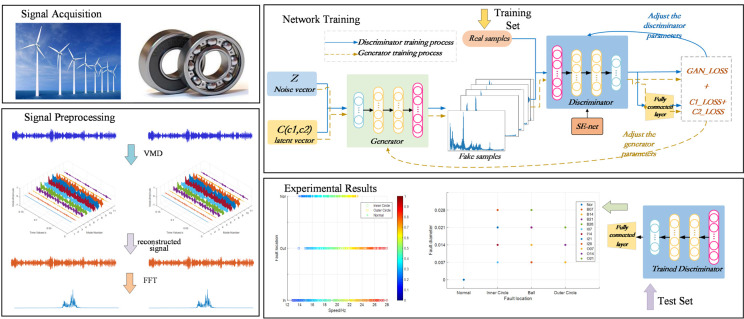
Model structure for interpretable bearing fault diagnosis.

**Figure 3 entropy-26-00480-f003:**
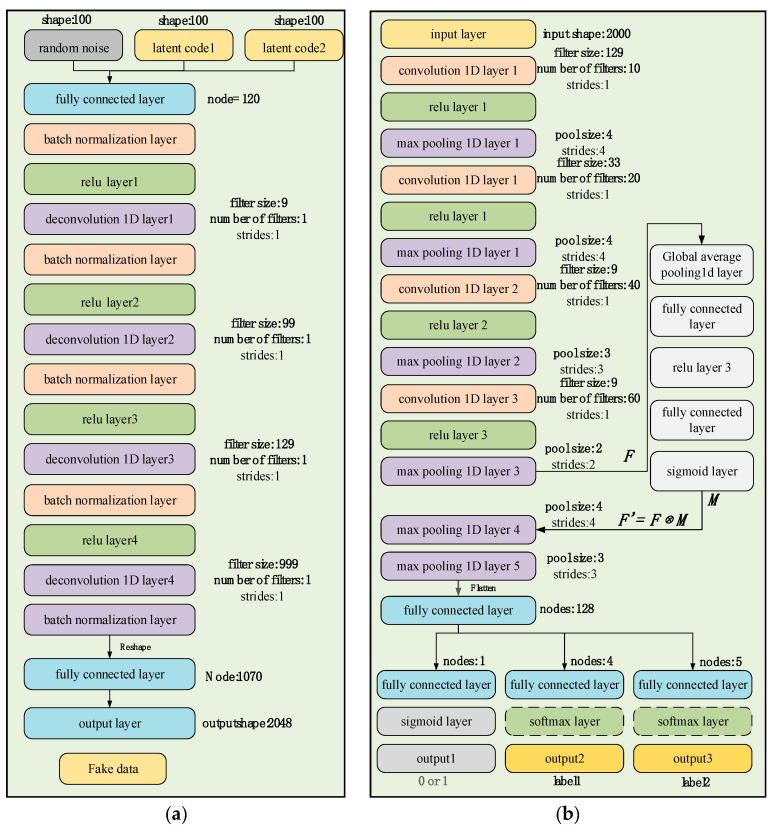
Network structure of generator and discriminator. (**a**) Generator, (**b**) Discriminator.

**Figure 4 entropy-26-00480-f004:**
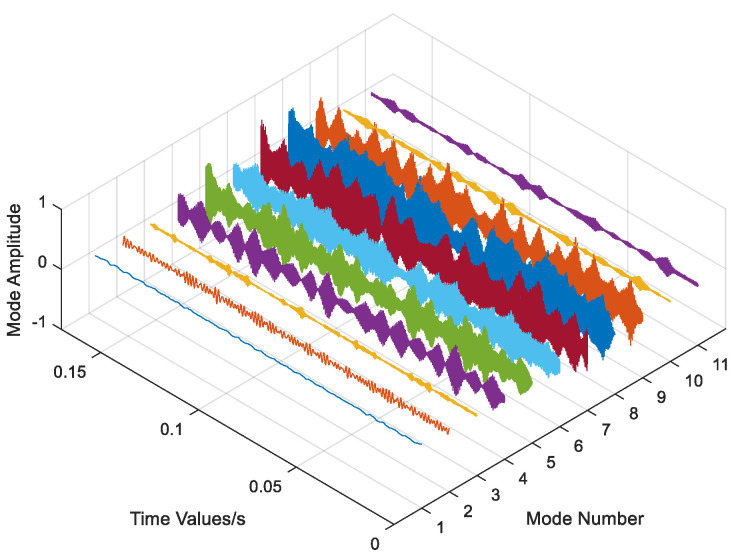
VMD decomposition result of Out07.

**Figure 5 entropy-26-00480-f005:**
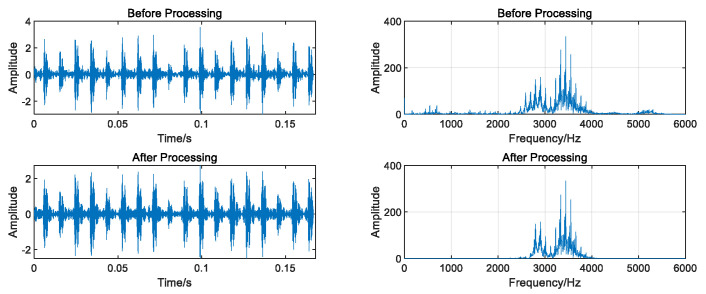
Comparison of time domain and frequency domain signal before and after processing.

**Figure 6 entropy-26-00480-f006:**
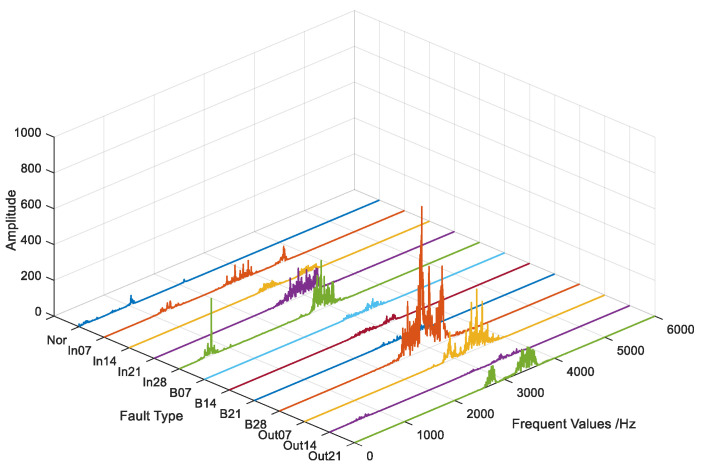
Comparison of 12 processed fault signals.

**Figure 7 entropy-26-00480-f007:**
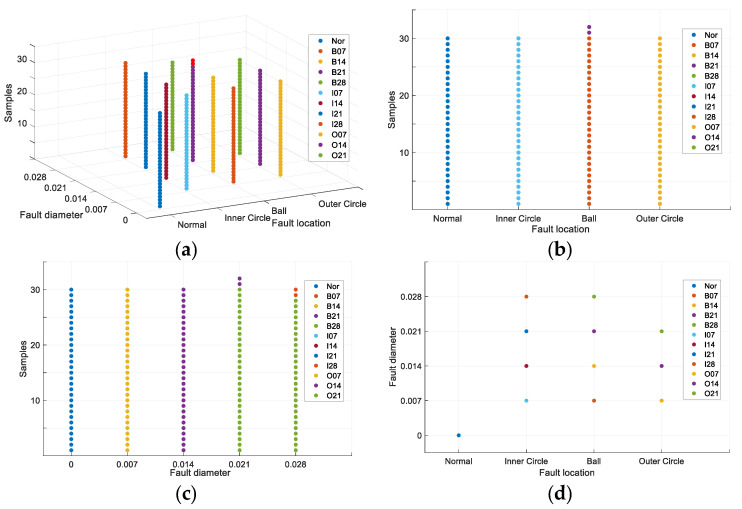
Interpretable diagnostic results for CRWU data set. (**a**) the three-dimensional diagram of the test set diagnosis. (**b**) the classification of the fault location. (**c**) the classification of the degree of damage. (**d**) an interpretable diagnosis of the two discrete features.

**Figure 8 entropy-26-00480-f008:**
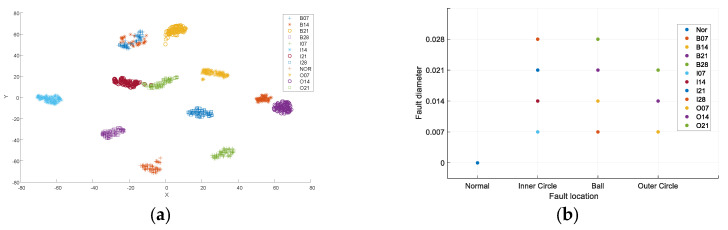
Comparison between the method in [23] and the interpretable diagnostic results. (**a**) the experimental effect diagram (**b**) the result of interpretable fault diagnosis.

**Figure 9 entropy-26-00480-f009:**
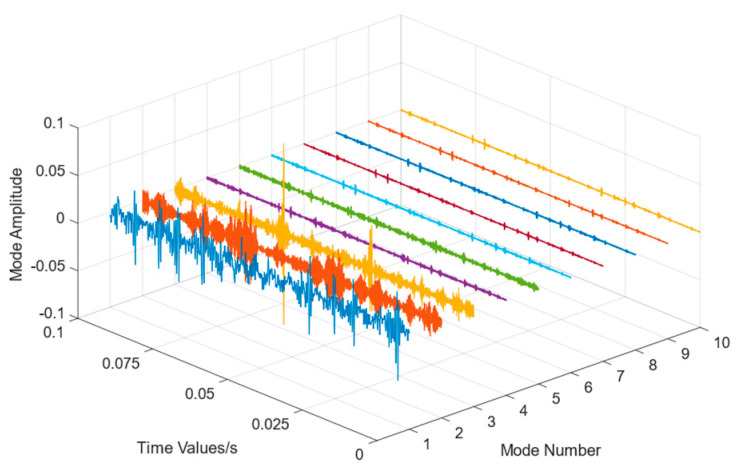
VMD decomposition result of inner circle fault bearing.

**Figure 10 entropy-26-00480-f010:**
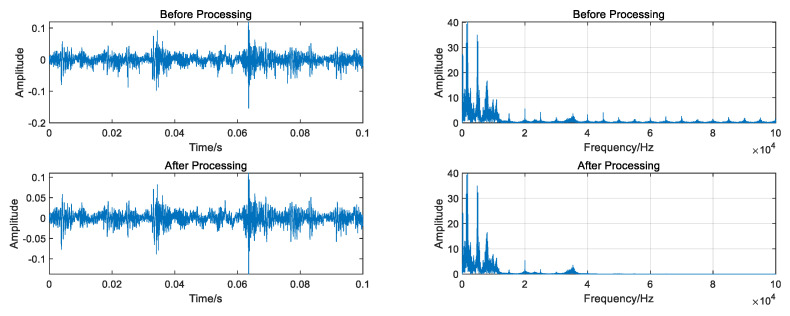
Comparison of time domain and frequency domain signal before and after processing.

**Figure 11 entropy-26-00480-f011:**
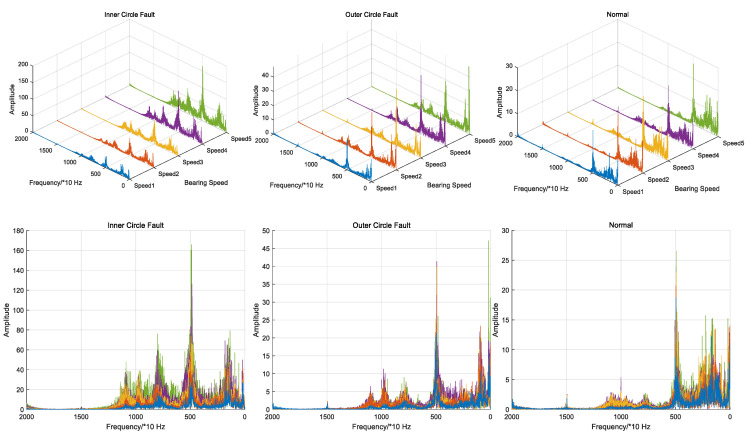
Frequency domain comparison of increasing speed signals.

**Figure 12 entropy-26-00480-f012:**
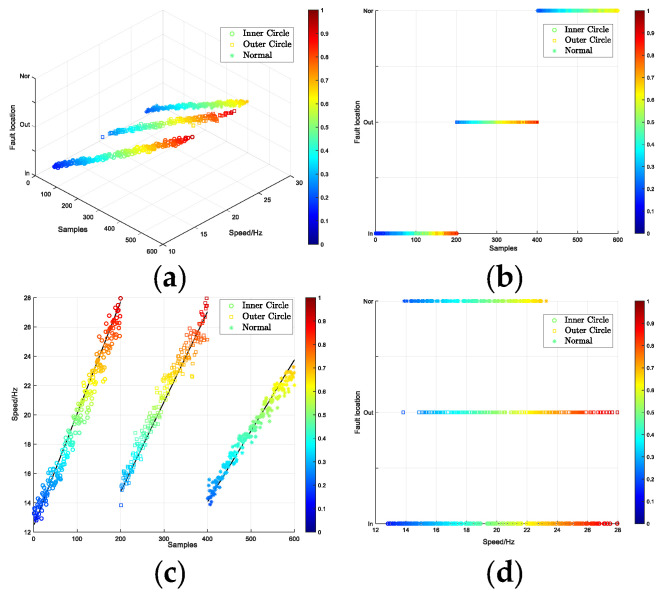
Interpretable diagnostic results under increasing speed. (**a**) the overview effect of the three-dimensional graph (**b**) the classification of the three fault locations (**c**) the diagnostic error display for speed (**d**) the interpretable diagnostic result of speed.

**Figure 13 entropy-26-00480-f013:**
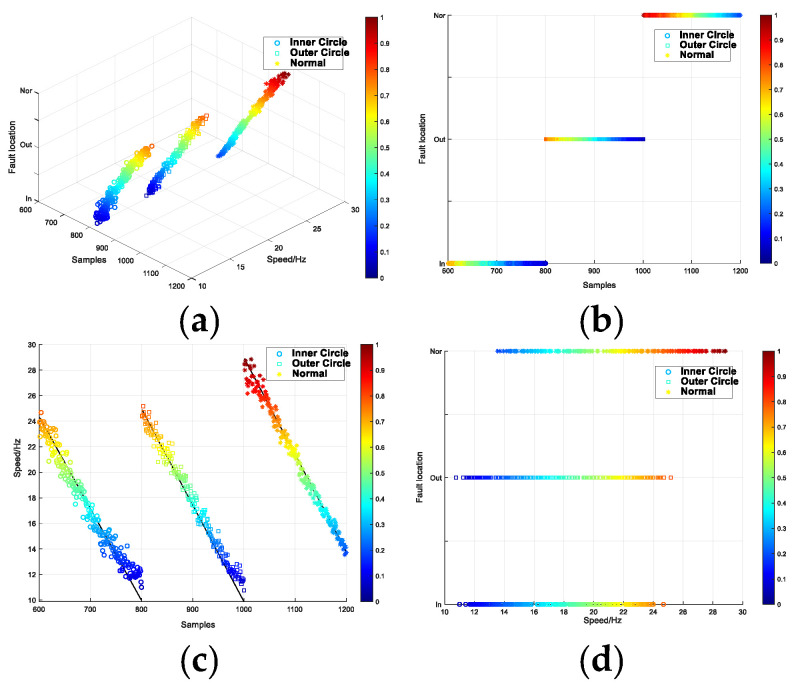
Interpretable diagnostic results under decreasing speed. (**a**): the overview effect of the three-dimensional graph (**b**): the classification of the three fault locations (**c**): the diagnostic error display for speed (**d**): the interpretable diagnostic result of speed.

**Figure 14 entropy-26-00480-f014:**
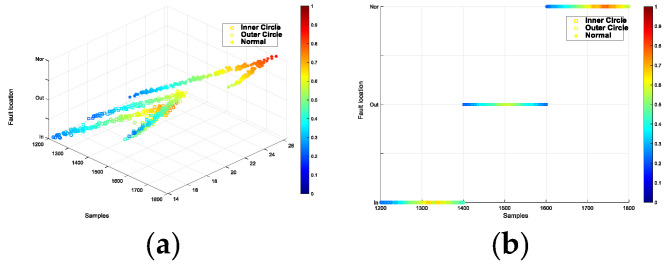
Interpretable diagnostic results under increasing then decreasing speed. (**a**) the overview effect of the three-dimensional graph (**b**) the classification of the three fault locations (**c**) the diagnostic error display for speed (**d**) the interpretable diagnostic result of speed.

**Figure 15 entropy-26-00480-f015:**
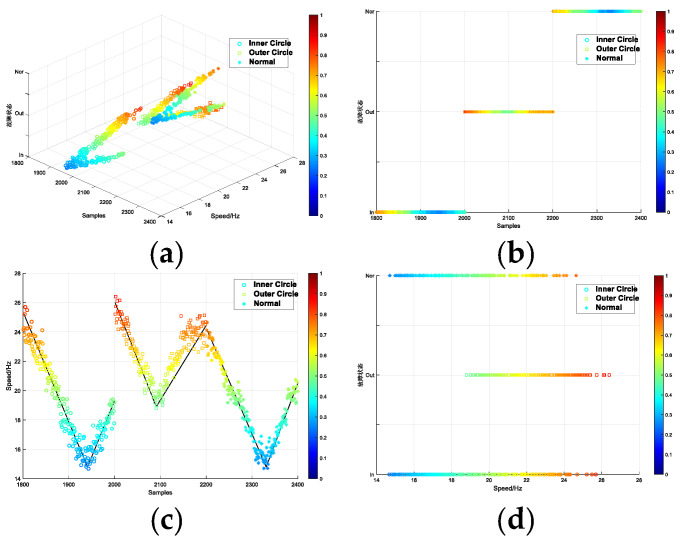
Interpretable diagnostic results under decreasing then increasing speed. (**a**) the overview effect of the three-dimensional graph (**b**) the classification of the three fault locations (**c**) the diagnostic error display for speed (**d**) the interpretable diagnostic result of speed.

**Figure 16 entropy-26-00480-f016:**
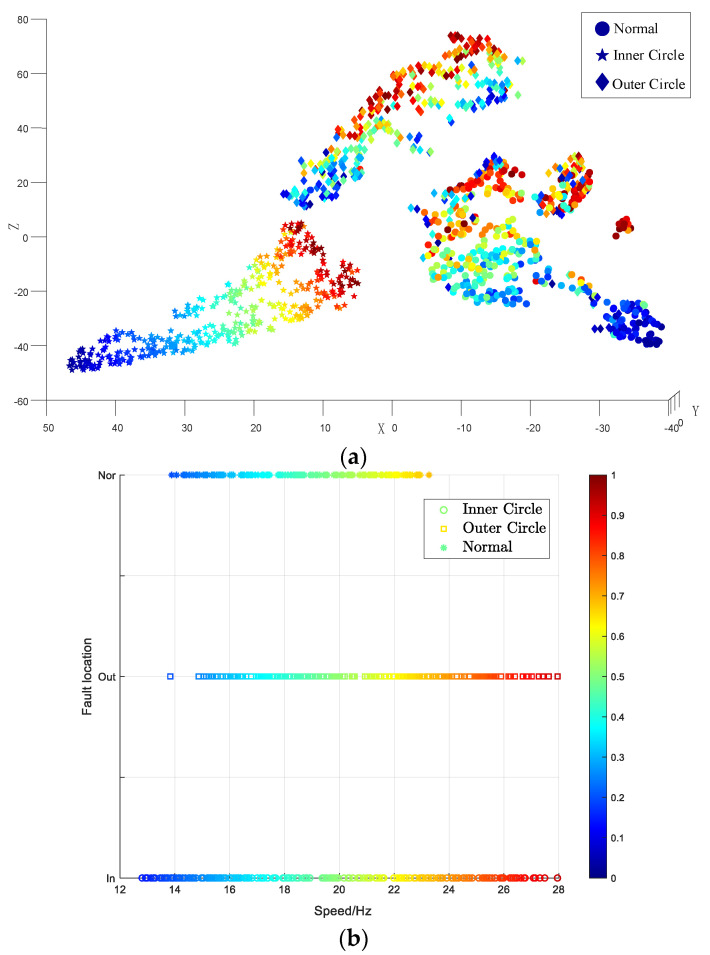
Comparison between the feature extraction method and the interpretable diagnostic results. (**a**) the feature extraction method (**b**)the interpretable diagnostic results.

**Figure 17 entropy-26-00480-f017:**
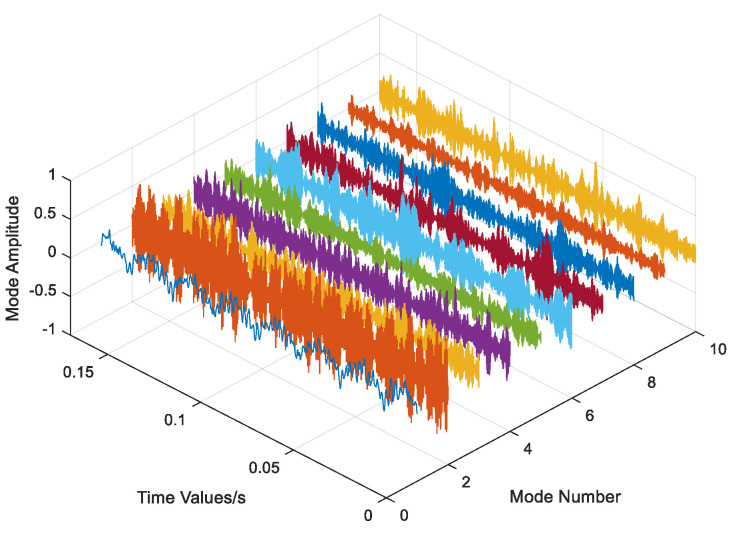
VMD decomposition result of the bearing signal in the first working condition.

**Figure 18 entropy-26-00480-f018:**
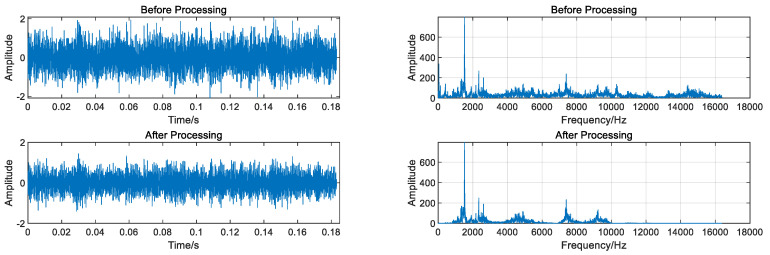
Comparison of time domain and frequency domain signal before and after processing.

**Figure 19 entropy-26-00480-f019:**
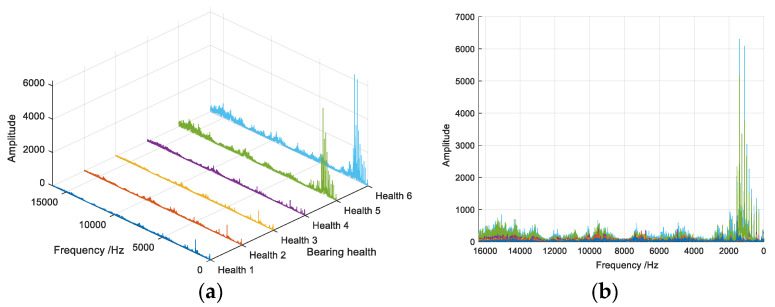
Signal frequency domain comparison of different health sections. (**a**) Three-dimensional comparison (**b**) Two-dimensional comparison.

**Figure 20 entropy-26-00480-f020:**
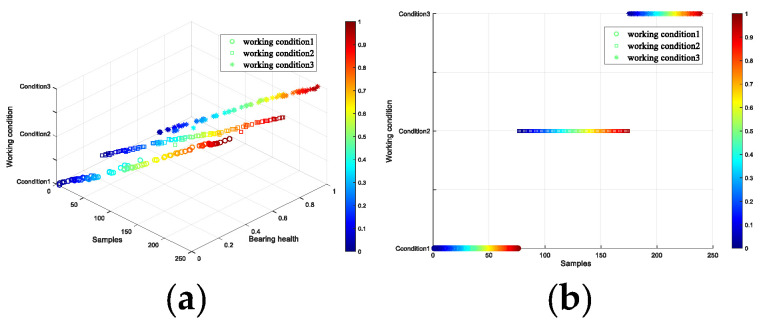
Interpretable diagnostic results of health and working conditions. (**a**) an overview of the three-dimensional graph (**b**) the classification of the three working conditions of the test set data (**c**) the diagnostic results for the health degree feature (**d**) the interpretable diagnostic results for the bearing data.

**Figure 21 entropy-26-00480-f021:**
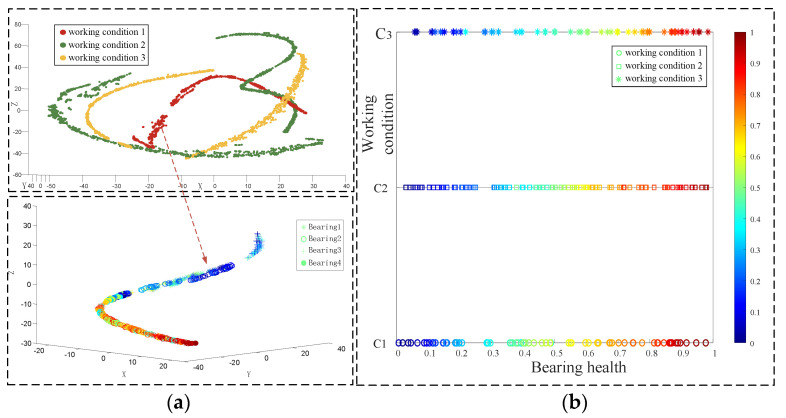
Interpretable diagnostic results of health and working conditions. (**a**) the vibration data under the three working conditions (**b**) the result obtained using the C-InGAN interpretable fault diagnosis model.

**Table 1 entropy-26-00480-t001:** Description of data from the CRWU data set.

Fault Location	Fault Diameter/in	c1	c2	Class Label
Normal	——	0	0	Nor
Inner circle	0.007	1	1	In07
	0.014	1	2	In14
	0.021	1	3	In21
	0.028	1	4	In28
Ball	0.007	2	1	B07
	0.014	2	2	B14
	0.021	2	3	B21
	0.028	2	4	B28
Outer circle	0.007	3	1	Out07
	0.014	3	2	Out14
	0.021	3	3	Out21

**Table 2 entropy-26-00480-t002:** Description of selected data from the Ottawa variable speed bearing dataset.

Fault Type	Speed Varying Conditions	c1	c2
Inner circle	Increasing speed	0	12.5–27.8
	Decreasing speed	0	24.3–9.9
	Increasing then decreasing speed	0	15.1–24.4–28.7
	Decreasing then increasing speed	0	25.3–14.8–19.4
Outer circle	Increasing speed	1	14.8–27.1
	Decreasing speed	1	24.9–9.8
	Increasing then decreasing speed	1	14.0–21.7–14.5
	Decreasing then increasing speed	1	26.0–18.9–24.5
Normal	Increasing speed	2	14.1–23.8
	Decreasing speed	2	28.9–13.7
	Increasing then decreasing speed	2	14.7–25.3–21.0
	Decreasing then increasing speed	2	24.2–14.8–20.6

**Table 3 entropy-26-00480-t003:** Description of selected data from Xi’an Jiaotong University accelerated degradation bearing data set.

Working Condition	Damage Location	Number of Samples	c1	c2
1	Outer circle	369	0	0-1
2	Outer circle	483	1	0-1
3	Outer circle	342	2	0-1

## Data Availability

Data are contained within the article.

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
