# Peer review of "Application of C-InGAN Model in Interpretable Feature of Bearing Fault Diagnosis"

_entropy, 2024, doi:10.3390/e26060480_

Round 1

Reviewer 1 Report

Comments and Suggestions for Authors

This paper introduces a Conditionally Interpretable Generative Adversarial Network (C-InGAN) for bearing fault diagnosis, emphasizing the model's effectiveness and interpretability of features demonstrated through experiments.

In summary, this manuscript has captured the interest of researchers in the relevant field. However, significant improvements are still needed for this manuscript. The following are my specific suggestions:

1. Expand the introduction to include a brief overview of existing methodologies and their limitations which your study addresses. This sets a solid foundation for understanding the novelty of your work.

2. Provide more detailed descriptions and justifications for the choices of the parameters in your C-InGAN model. Explain why specific configurations were chosen over others.

3. Results sections should be accompanied by more in-depth statistical analysis and interpretation. Discuss the implications of these results in the broader context of fault diagnosis.

4. Address the limitations of your study more comprehensively. Discuss potential biases, the scope of applicability of the model, and assumptions made during the research.

Comments on the Quality of English Language

The English level of the manuscript seems proficient, with a clear structure and technically appropriate language that is typical for academic publications in the field of engineering.

Reviewer 2 Report

Comments and Suggestions for Authors

Comments: To address the interpretable fault features in sample generation based on GAN, this paper proposed a new method named conditionally interpretable generative adversarial networks (C-InGAN). The Topic is very significant and interesting, with the effectiveness validated on three different datasets. Following are some comments to improve the readability.

1) What does the latent code specifically mean in this study, and how to determine its values? Please add more explanations.

2) In Eq.(3), how to quantify the Information I(c; D) between the latent code and generated data?

3)In line 300," ... was designed to aid the discriminator in completing the training more effectively", please add some results like the convergence speed to confirm such a conclusion.

4) How to determine the weights of different channels? and how do the values affect the discriminator's performance. 

5) As to the combination of both discrete and continuous features, you can refer to: 10.1109/TR.2022.3194107. Regarding to more interpretable structural parameters design for the network, you can refer to: https://doi.org/10.1016/j.aei.2023.101877.

Comments on the Quality of English Language

Overall, the English presentation is good and fluent, some small typos and format errors should be double-checked.
